# A pilot study of a single intermittent arm cycling exercise programme on people affected by Facioscapulohumeral dystrophy (FSHD)

**Fraser Philp**[ID][1]*, **Richa Kulshrestha**[2], **Nicholas Emery**[2], **Marco Arkesteijn**[3], **Anand Pandyan**[4], **Tracey Willis**[2,5]

1 School of Health Sciences, University of Liverpool, Liverpool, United Kingdom, 2 Robert Jones and Agnes Hunt Orthopaedic Hospital (RJAH) Foundation Trust, Gobowen, Oswestry, United Kingdom, 3 Institute of Biological, Environmental & Rural Sciences, Aberystwyth University, Plas Gogerddan, United Kingdom, 4 Faculty of Health and Social Sciences, Bournemouth University, Poole, United Kingdom, 5 Chester Medical School, University of Chester, Chester, United Kingdom

* f.philp@liverpool.ac.uk

**Data Availability Statement:** Our ethics approval does not allow for our study data to be stored on an open access data repository. Due to ethical

## Abstract

For patients affected by Facioscapulohumeral dystrophy (FSHD), alternate methods for increasing physical activity engagement that may benefit shoulder function and wider health are needed. Arm cycling has been proposed as a potential method for achieving this although dosage parameters and evidence is limited. The aim of this study was to conduct a pilot study evaluating the effect of a single intermittent arm cycling exercise programme on people affected by FSHD. People with confirmed genetic diagnosis of FSHD between the ages 18–60 years were recruited to attend a single session for the exercise intervention (5 exercise efforts lasting 2 minutes each with 30 seconds of rest between each effort). Prior to exercise, measures of shoulder function (Oxford shoulder score), strength and range of movement were recorded. During the exercise participants were video recorded to quantify range of movement and extract movement profile features. Participants comments were recorded and followed up four days later to check for adverse events. Fifteen participants, (6F:9M) were recruited with median (IQR) Oxford Shoulder Scores of 25 (18 to 39). All participants successfully completed the exercise intervention with only transient symptoms consistent with exercise being reported and achieving a median (IQR) rate of perceived exertion scores of 13 (12 to 13). Movement profile data was available for 12 out of 15 participants and suggests that exercise intensity did not compromise movement. An association between strength and shoulder function ($R^2 = 0.5147$), Rate of perceived exertion (RPE) of the final effort against shoulder function and strength ($R^2 = 0.2344$ and $0.1743$ respectively) was identified. Participant comments were positive regarding the exercise intervention. Our study demonstrates that an intermittent arm cycling programme is feasible for people affected by FSHD. Further work is needed to evaluate physiological responses to exercise across variations in programme variables and equipment set up in a larger sample of people affected by FSHD.

restrictions, the data for this study is only available on request from rjah.researchoffice@nhs.net, to researchers who can demonstrate legitimate interest and appropriate ethical approval.

**Funding:** The study was funded by The Orthopaedic Institute Limited grant reference number RPG162. The funders had no role in study design, data collection and analysis, decision to publish, or preparation of the manuscript

**Competing interests:** I have read the journal's policy and the authors of this manuscript have the following competing interests: I am reporting that Professor Anand Pandyan has received unrestricted educational support from Allergan and Biometrics Ltd., and Honorarium payments from Allergan, Biometrics Ltd, Ipsen, and Merz. These companies are unlikely to be affected by the research reported in the enclosed paper. I have disclosed those interests fully and I have in place an approved plan for managing any potential conflicts arising from that involvement. This does not alter our adherence to PLOS ONE policies on sharing data and materials.

## Introduction

Facioscapulohumeral dystrophy (FSHD) is a muscular dystrophy resulting from an altered contraction on chromosome 4 and disrupted methylation allowing for DUX4 transcription [1, 2]. Several signalling pathways in skeletal muscle are affected, resulting in increased oxidative stress, downregulation of myogenesis and ultimately cell death [2]. Body structure and function is affected with notable degeneration of the muscle fibres, characterised by fatty and fibrotic muscle infiltration and increased fibre size variability [2, 3]. FSHD most frequently affects muscles of the upper-limbs and torso, although the lower limbs may also be affected [4]. FSHD is non-life limiting with a relatively slow but asymmetric and variable disease progression [4]. People affected by FSHD present with functional impairments of decreased muscle mass strength and shoulder mobility [4–6]. As a result upper-limb function, particularly in activities requiring shoulder elevation are negatively affected [4]. Secondary complications of FSHD can also include chronic pain, fatigue and in some cases, decreased mobility and pulmonary function which negatively impact quality of life [4, 5, 7].

Exercise based and physical activity is the primary form of rehabilitation in people affected by FSHD. Most physical activity strategies in people affected by neuromuscular conditions, including FSHD, focus on the lower limb and address exercise components related to weight-bearing, functional, stretching and sensor-based exercise programmes [8]. It has been identified that existing strategies and methods used to elicit increases in physical activity are not explicit [8]. Most studies do not meet the minimum requirements recommended for strength training [9–11], whilst only one study has met the requirements recommended for physical activity related to cardiovascular respiratory fitness [11]. It is important to acknowledge that the guidelines used to determine thresholds/targets for physical activity are not specific to neuromuscular diseases and have been based on normative datasets of people without pathology. Their validity in acting as bench marks for determining exercise prescription and physical activity parameters in people affected by neuromuscular diseases requires further investigation.

The overall effectiveness of exercise interventions for increasing physical activity in people with FSHD and neuromuscular disorders is uncertain [8]. It appears that existing exercise interventions may positively influence overall physical fitness, work capacity, quality of life and fatigue resistance, whilst having limited influence on function, strength and pain [8]. However, this needs to be considered alongside the methodological limitations of the studies on which these conclusions are based [8, 11]. Some studies have shown that exercise with moderate weights or resistance is not detrimental to people with FSHD [12, 13] and that consistent aerobic training in people with FSHD may improve improves cardiovascular fitness and strength concomitantly [14]. Whilst adverse events are unlikely when people with neuromuscular diseases undertake exercise at moderate intensities, there are still possible dependent on the protocol and subject [11, 15].

There is therefore limited evidence available to inform clinical practice and support strategies for improving physical activity and exercise for people affected by FSHD [8]. This includes identification of measures e.g. body structure (e.g. movement features) or body systems responses (e.g. cardiovascular) that can be used to identify fatigue leading to adverse exercise responses or monitoring of changes in control during exercise activities. Given the limited functional capacity and secondary complications experienced in people affected by FSHD, there is a need identify alternate methods for increasing and measuring physical activity engagement that may benefit shoulder function and wider health. This is important as a majority of the impairments in patients with FSHD occur in the upper-limb, limiting activities dependant on upper-limb function and participation. Arm cycling has been proposed as a

potential method for achieving this through improvement of muscle strength and aerobic capacity, although dosage parameters and evidence is limited. Assisted arm cycling has been used successfully in young people with Duchenne Muscular Dystrophy [16], but only a single case study of a person severely affected by FSHD has been conducted with no improvement in functional assessments [17]. Before arm cycling can be used in people affected by FSHD, further work is needed to evaluate the overall feasibility. The aim of this study was therefore to conduct a pilot study evaluating the effect of a single intermittent arm cycling exercise programme on people affected by FSHD.

## Materials and methods

Ethical approval for this study has been approved by North West—Greater Manchester East Research Ethics Committee reference:16/NW/0673 and conforms to Helsinki Declaration. This trial is registered on ClinicalTrials.gov Identifier: NCT04267354 available at https:// clinicaltrials.gov/ct2/show/NCT04267354. An overview of the study processes has been outlined in Fig 1. During the study there were no deviations to the protocol.

### Recruitment

Participants were recruited over a 12-month period between January 2017 to December 2017. Participants were recruited from an existing clinical database associated with the study site, the UK FSHD Patient Registry [18] and FSHD charity [19] using convenient sampling. Patients with confirmed genetic diagnosis of FSHD between the ages 18–60 years were included. Participants had to be willing to attend the assessment session and able to understand the participant information sheet and provide written informed consent. Participants with previous shoulder trauma or co-morbidities that would affect their ability to perform arm cycling were excluded. Participants were required to attend a single session for the exercise intervention and recording of measures relating to shoulder and elbow function, strength and range of movement. Participants were also followed up with a phone call four days later to record comments pertaining to their experience of the intervention and investigate any adverse effects. A sample size of 15 participants was selected as a pragmatic solution to the lack of evidence regarding adverse events to exercise, estimated recruitment rates and absence of agreed and validated primary outcome measures in this patient group.

### Outcome measures

Shoulder function was assessed using the Oxford Shoulder Score [20]. Passive Range of Movement for the bilateral shoulder and elbow joints was conducted using a goniometer and maximum bilateral shoulder and elbow strength values over three attempts were recorded using a hand-held dynamometer (CITEC HHD CT3002) and a 'break test' approach. The maximum value of three attempts was used in the analysis.

During the arm cycling exercise intervention, participants rating of perceived exertion ('effort') during arm cycling using the modified Borg Rate of Perceived Exertion (RPE) scale for each of the exercise attempts was recorded [21]. Participant comments about subjective description of ease of doing arm cycling was also noted. Video analysis of the participants was performed during arm cycling in the sagittal view, with a 3D single camera-based system, Codamotion, Charnwood Dynamics Ltd., Leicestershire, UK at 100Hz allowing for measurement of shoulder and elbow flexion/extension values. This was done to explore if any features of fatigue or adverse responses could be identified based on participant movement profile characteristics. Markers were placed on the distal head of fifth metacarpal bone, styloid process of ulna, lateral epicondyle of humerus and lateral aspect of acromion process and most lateral

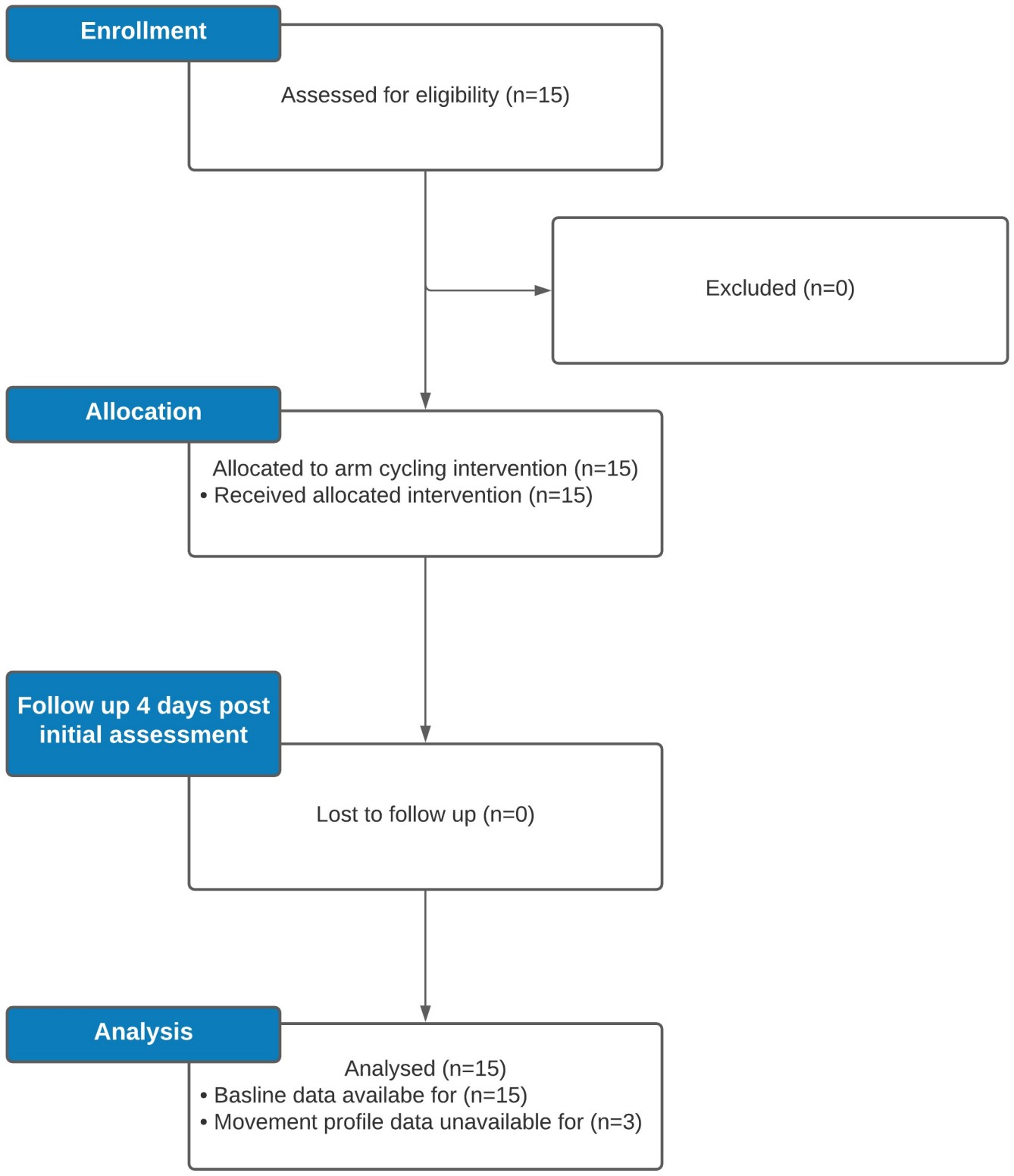

**Fig 1. Flowchart outlining study processes.**

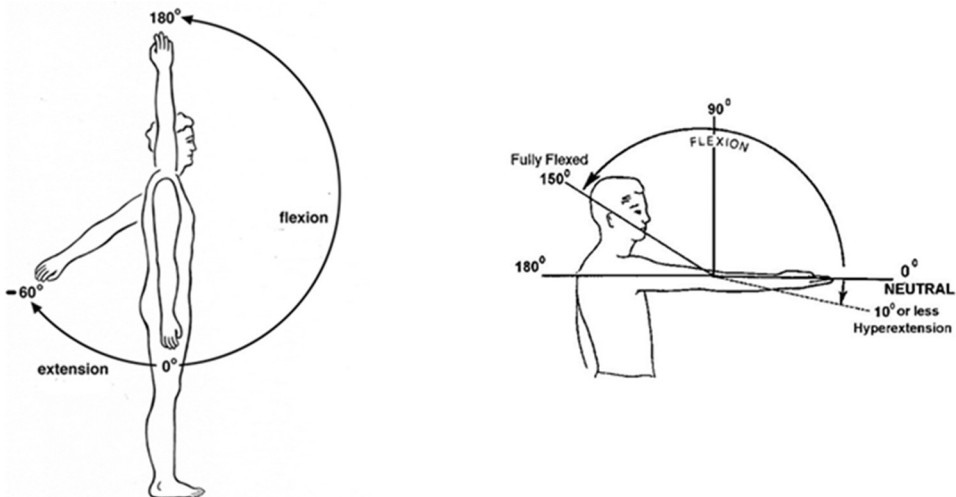

**Fig 2. Definition of joint angles.** For the shoulder, full flexion was defined as 180˚, and 0˚ was defined with the arm alongside the body, and extension was defined as negative values. For the elbow, full flexion was defined as 180˚, and full extension as 0˚.

part of hip on the side. These were digitized and joint angles calculated from the lines defining each respective segment between these points. The marker on hip was placed for indication of trunk orientation. Angles conventions were defined according to Fig 2.

### Arm cycling exercise intervention

**Equipment and set up.** Participants performed arm cycling using a table-top arm cycle and were overseen by a physiotherapist. The arm cycle was placed on table with height adjusted to be the same as the acromion (highest point on shoulder).

**Arm cycling exercise protocol.** Prior to exercise participants were familiarized on machine followed by 5 minutes rest prior to commencing arm cycling. Participants completed 5 exercise efforts lasting 2 minutes each with 30 seconds of rest between each effort, reflecting an intermittent exercise programme. This exercise programme type was selected as it allows for distinct periods of exercise efforts, between which, patients' responses to exercise could be evaluated i.e. development of delayed or immediate adverse responses to exercise. As there are no guidelines specific to this exercise intervention in patients with FSHD, the intermittent programme type was a pragmatic selection allowing for consideration of patient safety and evaluation of a dose-response relationship to exercise. Participants started the first exercise effort at the lowest resistance and a self-selected cadence. Participants were able to increase or decrease their selected level of resistance during any of the efforts which was recorded by the physiotherapist. People affected by muscular dystrophies are advised to work between 3 to 5 (moderate to hard) on the Modified Borg scale which is equivalent to 10 to 12 on the Original Borg scale (Fairly light to somewhat hard) [10, 21–23]. An RPE range of 11 to 13 (3.0–5.9 METs) is recommended for physical activity [23].

### Analysis

### Patient demographics and assessment measures

For patient demographics, including shoulder function, shoulder and elbow passive range of movement and maximum strength values, descriptive statistical analysis i.e. frequencies, median (IQR) were reported.

### Kinematic data

For kinematic data, a 25 to 30 second sample from the first and final exercise efforts i.e. efforts one and five, were evaluated for comparison and identification of movement profiles. For active range of movement at the shoulder and elbow, the median (IQR) minimum, maximum and overall range values are reported. The cadence and RPE of each respective attempt were also reported in this way. Movement profiles were evaluated by calculating time lag (mean difference in time between peak elbow and shoulder joint angles), mean phase angle values, shoulder and elbow joint velocities and the cross-correlation between shoulder and elbow joint movements in MATLAB V.2018A (Math-Works, Massachusetts, USA). Individual shoulder and elbow angle-angle plots comparing the first and final exercise effort attempts for each individual participant were created.

### Relationship between shoulder function, strength and effort/performance

To investigate the relationship between function and performance, shoulder function, (Oxford Shoulder Score) was plotted against maximum shoulder strength (Right shoulder flexion) and RPE of the final exercise effort independently. Pearson's correlations for measures of maximum strength at the joints and rationale for selection of right maximum shoulder strength are available in S1 Fig and were analysed using SPSS (IBM Corp. Released 2020. IBM SPSS Statistics for Windows, Version 27.0. Armonk, NY: IBM Corp). Maximum shoulder strength was also plotted against RPE of the final exercise effort.

### Patient comments

A summary of the noted participant comments and quantification of main features expressed, both during the exercise intervention session and four days later, have been presented.

## Results

### Results for patient demographics and assessment measures

Fifteen participants, (6F:9M) were recruited for this study. The median (IQR) age of participants was 45 (38 to 45). The median (IQR) Oxford Shoulder Score was 25 (18 to 39). Baseline median passive range of movement and maximum strength values for the shoulder and elbow joints have been reported in Table 1. All participants successfully completed the exercise intervention.

**Table 1. Baseline median passive range of movement and maximum strength values for the shoulder and elbow joints.**

| Joint | Shoulder | | | | | | Elbow | | | |
|---|---|---|---|---|---|---|---|---|---|---|
| Movement | Flexion | | Extension | | Abduction | | Flexion | | Extension | |
| Side | *Right* | *Left* | *Right* | *Left* | *Right* | *Left* | *Right* | *Left* | *Right* | *Left* |
| ***Median PROM (IQR) (°)*** | 100 (65–130) | 100 (80–130) | 55 (40–65) | 45 (35–70) | 110 (85–135) | 105 (90–130) | 143 (140–150) | 145 (140–150) | 0 (0–5) | 0 (0–5) |
| ***Median Maximum strength (IQR) (kgf)*** | 74 (55–130) | 76 (54–126) | 68 (40–85) | 70 (39–152) | 75 (67–165) | 72 (69–136) | 77 (28–134) | 52 (35–108) | 62 (31–127) | 54 (18–150) |

PROM = passive range of movement

### Results for kinematic data

Kinematic data were available for 12 out of the 15 participants. The data from 3 participants was unusable owing to insufficient marker tracking required for calculating joint angles. Time lag values ranged from -1.6 to 0.9 seconds and the mean phase angle values ranged from 28.9˚ to 38.0˚. For time lag values, the elbow typically led the shoulder, apart from five participants efforts where shoulder led the elbow and this varied between the first and last effort. The median (IQR) signal correlation value was 0.7 (0.6 to 0.7). Angle-Angle plots for shoulder and elbow function are presented in Fig 3.

For most participants there was synchronicity between shoulder and elbow movements. With participants 10, and 15 demonstrating the least synchronicity.

During the arm cycling exercise intervention, median (IQR) maximum shoulder and elbow velocities were 224.5˚/s (167.8˚/s to 613.1˚/s) and 468.9˚/s (270.7˚/s to 1071.5˚/s) respectively. The median (IQR) for minimum active range of movement values for shoulder and elbow flexion/extension values were 105.6˚ (101.0˚ to 108.8˚) and 39.9 (34.4˚ to 50.9˚) respectively. The median (IQR) for maximum active range of movement values for shoulder and elbow flexion/extension values were 151.0˚ (145.5˚ to 159.3˚) and 123.6˚ (116.5˚ to 128.2˚) respectively. The median (IQR) range values for active range of movement at the shoulder and elbow for flexion/extension values were 44.1˚ (39.2˚ to 54.1˚) and 80.5˚ (76.5˚ to 86.5˚) respectively.

The median cadence (IQR) for the first and final exercise efforts were 60 (44 to 68) and 68 (53 to 76) respectively. RPE median (IQR) scores were 11 (9 to 11) and 13 (12 to 13) for the first and final exercise attempts respectively. An overview the progression/regressions in resistance for each of the five exercise efforts are presented in Fig 4.

### Results for relationship between shoulder function, strength and effort/ performance

Maximum shoulder strength was able to account for some of the variability observed in shoulder function, with the highest $R^2$ value of 0.5147 (Fig 5). Whilst RPE of the final effort was able

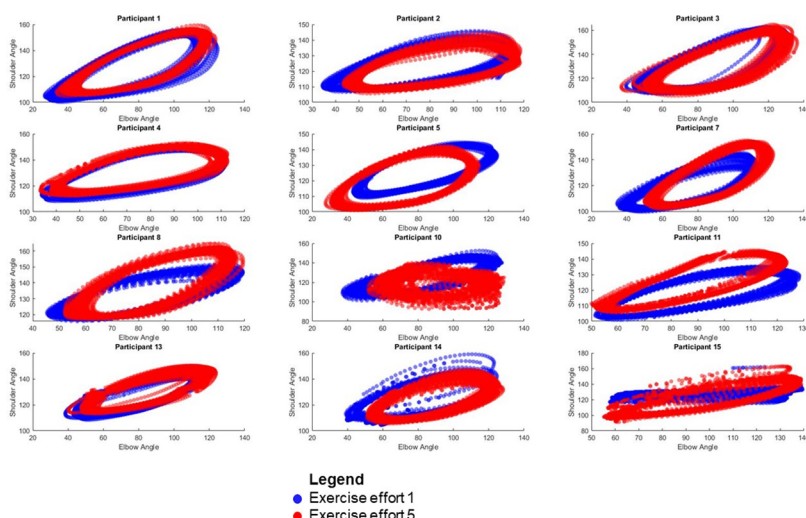

**Fig 3. Shoulder and elbow angle-angle plots for the first and last exercise effort.** * gaps/ breaks indicate missing data.

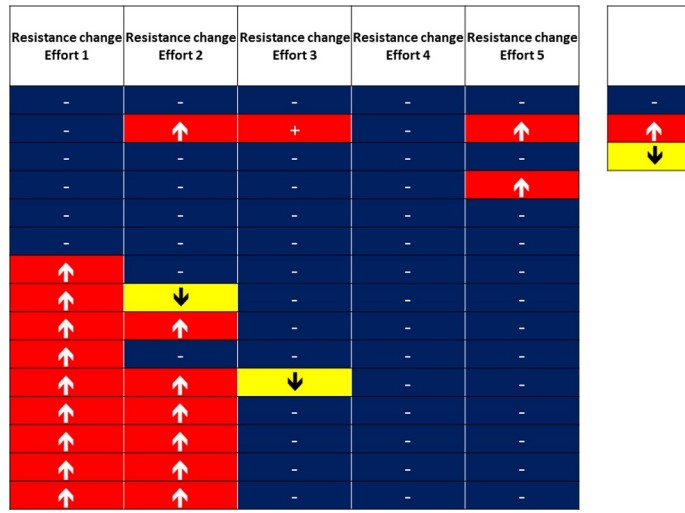

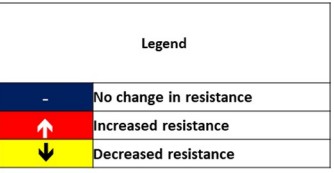

**Fig 4. Individual progression/regressions in resistance for each of the five exercise efforts.** Eleven out of 15 participants (73%) selected to increase their resistance over the five exercise efforts.

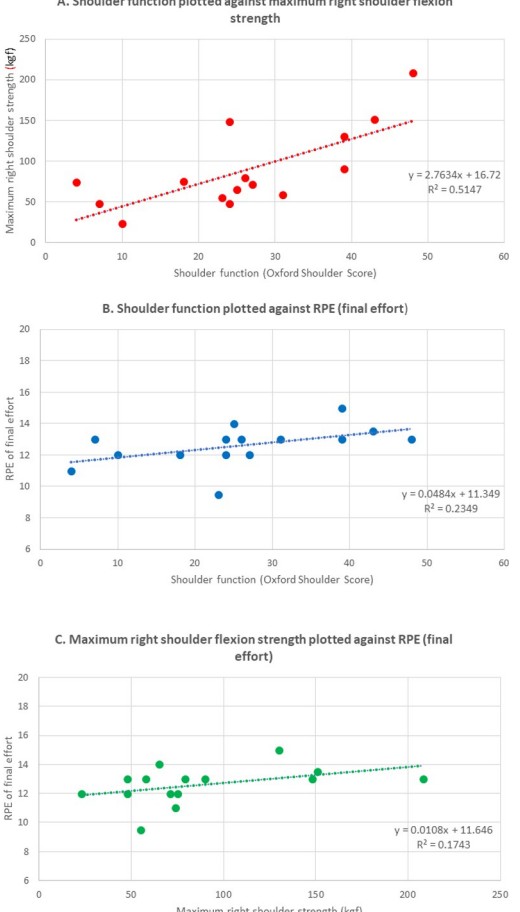

**Fig 5.** Graphs of A. Shoulder function plotted against strength, B. Shoulder function plotted against RPE and C. Strength plotted against RPE. $R^2$ values represent the coefficient of determination.

to account for some of the variability observed in shoulder function and strength, this was limited with $R^2$ values of 0.2344 and 0.1743 respectively.

## Results for participants comments about their experience of the exercise intervention

During the exercise intervention no participants reported any significant adverse events and this was also true when followed up four days later. During the exercise session participants reported a range of transient symptoms consistent with exercise, namely, upper limb ache (n = 5, 33%), upper limb heaviness, (n = 5, 33%), upper limb fatigue (n = 5, 33%), upper limb muscular effort (n = 3, 20%), increased respiratory work (n = 2, 13%), tingling (n = 2, 13%), warmth (n = 2, 13%), tightness, stiffness and weakness (n = 1, 7%). At the four day follow up seven out 15 participants reported experiencing transient stiffness (n = 4, 27%), ache (n = 2, 13%) and heaviness (n = 1, 7%) after the exercise session which lasted until the following day. Overall participant comments were positive when asked about their experience of the intervention with 13 out of 15 participants (87%) expressing they would consider arm cycling as a part of their routine rehabilitation if shown to be effective. Positive comments also extended to the ability of people to engage with the exercise intervention and associated equipment in their home environment. The arm cycling intervention was identified as convenient with regards to the equipment and storage and overall exercise protocol. Some limitations regarding setting up equipment at home to correct height (n = 2, 13%) continued motivation and scheduling interventions into daily routine (n = 3, 20%), were identified.

## Discussion

### Feasibility of arm cycling exercise

The aim of this study was to conduct a pilot study evaluating the effect of a single intermittent arm cycling exercise programme on people affected by FSHD. We have demonstrated that arm cycling, comprised of five exercise efforts lasting two minutes at a self-selected intensity with 30 seconds of rest, is feasible for people affected by FSHD, including those with low levels of shoulder function. The ability to complete arm cycling successfully extends previous studies that have reported that arm cycling was feasible during assisted arm cycling in both people with FSHD (n = 1) and Duchenne Muscular Dystrophy [16, 17]. Whilst participants did report symptoms such as ache, heaviness and fatigue, these resolved the following day and overall are consistent with responses to exercise and findings of similar studies [8, 24]. Participant comments regarding the intervention were mainly positive and suggest that if demonstrated to be effective, participants would consider arm cycling as a part of their rehabilitation / routine physical activity. Further work is needed to investigate the physiological response of upper-limb exercise, including arm cycling, in a larger sample of people affected by FSHD.

### Exercise intensity achieved during session

Nearly all participants (14 out of 15) were able to perform the exercise at the minimum recommended RPE intensities during any of the exercise efforts. RPE median (IQR) scores were 11 (9 to 11) and 13 (12 to 13) for the first and final exercise attempts respectively. A broad range of RPE thresholds (range 12 to 19) have been used to inform cardiovascular and strengthening exercises in the literature, predominantly for the lower limb and for other similar neuromuscular conditions [25]. It is important to recognise that the relationship between Borg scale values and desired physiological responses to exercise are based on data of people without pathology [23]. Previous research has shown that there may be an offset or non-linear change

in this assumed relationship at different intensities and for people with FSHD or other acquired neurological diseases with shared similar symptoms such as fatigue [26, 27]. Further work is needed to evaluate the physiological responses to arm cycling and develop guidelines comprised of quantified ranges that can for exercise prescription. This should be done to ensure participants are achieving sufficient physiological overload and prevent adverse responses to exercise. Further work is also needed to explore the longitudinal responses to the exercise intervention over repeated efforts and a period of time. Suitable exercise testing protocols and responses to different combinations of exercise prescription variables also require further investigation. These should be considered alongside other measures of psychological wellbeing and participation.

The movement profiles (12 out of a possible 15 participants), demonstrate consistent synchronised movement. Given that movement was not compromised we surmise that fatigue may not have occurred. This was supported by the small-time lag values, large signal correlation values, mean phase angle values and synchronous shoulder and elbow angle-angle plots apart from two (participants 10 and 15). There appears to be agreement between the kinematic variables /movement profiles of participants and RPE scores recorded, however, these results need to be interpreted against the previously identified limitations. Whilst the exercise protocol did not appear to compromise the functional capacity of participants, it is also possible that participants were not exercising at an intensity or durations required to achieve physiological overload. The less synchronous shoulder and elbow angle-angle plots associated with participants 10 and 15 may possibly suggest that they had reached a higher level of fatigue or were using alternate movement strategies to engage with the activity. These assumptions are however based on crude metrics of performance and so further work is needed to develop sensitive measures of fatigue which can be used to inform monitoring for exercise progression and regression. No other explanatory factors explaining the profiles of participants 10 and 15 were evident when evaluating the dataset.

## Considerations associated with exercise protocol

Additional strategies for optimising the arm cycling intervention so that the shoulder joint or other muscles are differentially loaded are required. Participants primarily drove the arm cycling protocol by leading with their elbow movements indicated by the time lag values. This is likely reflective of the equipment set up and task requirements but may also be influenced by the functional capacity of people affected by FSHD. This is important given the heterogenous presentation of FSHD and loss of shoulder movement associated with the condition [4]. An association between strength and shoulder function, RPE of the final effort against shoulder function and strength was identified. This possibly suggests that participants with higher levels of function can undertake arm cycling at a higher intensity. A limitation of our study was that we were unable to quantify the variation in load between participants. The equipment used in this study was selected given the overall cost and ease of use which likely increases overall accessibility to people affected by FSHD, allowing engagement with physical activity. However, a better understanding of the overall load associated with the exercise and quantified performance of people affected by FSHD will allow for better informed exercise assessment and prescription development and monitoring. Equipment set up and exercise protocol likely accounts for the differences in active range of movement and cadence values observed in our study when compared to people without pathology and other patient groups [28–30]. It is important to understand the effect of these variation on physiological responses to exercise. Challenges regarding identical equipment set, specifically height, in a community setting was identified by two participants. The transferability of arm cycling for increasing physical activity

in a community setting and physiological responses to exercise as a result of variability therefore need to be better evaluated.

The overall duration of time spent engaging in exercise that could be considered physical activity in our study was limited i.e. 10 minutes. However upper-limb rehabilitation and strategies for increasing physical activity is not well studied in this population and there is no evidence of the effects of arm cycling on cardiorespiratory fitness or muscle function. Previous studies have shown cardio-respiratory fitness improvements in people affected by neuromuscular disease but these have used lower limb static exercise bikes [31, 32]. Similarly, to guidelines for strength training, recommendations for cardiovascular fitness i.e. moderate intensity exercise for 150 min.wk-1, are based on people without pathology in order to prevent complications associated with insufficient activity such as cardiovascular disease [33, 34]. Given that people affected by FSHD may experience secondary complications associated with the disease such as chronic pain and fatigue [5, 35], existing exercise protocols and exercise guidelines need to consider these factors to maximise engagement and minimise risk of harm. Alternate exercise programme variables (e.g. intensity, time, body structure (lower versus upper limb)) may be used to achieve specific or concurrent responses to exercise (e.g. endurance, strength, general physical activity) on the basis of first principle arguments or clinical reasoning, which maybe reflective of current clinical practice. However, these need to be considered against the previously discussed points.

The arm cycling intervention was identified as convenient both in terms of equipment and overall protocol. However continued motivation and scheduling interventions into daily routine (n = 3, 20%), were identified as possible barriers by participants. It is recognised that there are additional factors which may influence engagement and maintenance of physical activity interventions such as availability of appropriate information, accessibility to equipment and space, cost and time [8, 24, 36]. Future work should therefore look to address potential barriers to engagement and maintenance of physical activity interventions such as arm cycling which may be facilitated by technological solutions.

## Conclusion

A single episode of arm cycling, comprised of five exercise efforts lasting two minutes at a self-selected intensity with 30 seconds of rest, is feasible for people affected by FSHD. No significant long-term adverse events were reported as a result of the exercise protocol. Further work is needed to evaluate physiological responses to exercise across variations in programme variables and equipment set up in a larger sample of people affected by FSHD. Barriers to undertaking and maintaining engagement with arm cycling in the community need to be considered in the design of future studies. Future work should also look to identify sensitive measures and outcomes that can be used to prescribe exercise at the appropriate intensity and identify fatigue, preventing adverse responses.

## Supporting information

**S1 Fig. Correlation matrix for maximum strength values at shoulder and elbow joints.**
Right maximum shoulder flexion strength was found to be highly correlated (range 0.725 to 0.957) with all other measures of shoulder and elbow strength. Selection of strength in this plane was also supported by the fact that people affected by FSHD often lose functional overhead movement in this plane.
(TIF)

**S1 File. TREND statement checklist.**
(PDF)

**S2 File. Approved study protocol.**
(PDF)

## Acknowledgments

Our sincere thanks to UK FSHD Patient Registry that helped us to recruit people affected by FSHD for the study.

## Author Contributions

**Conceptualization:** Richa Kulshrestha, Nicholas Emery, Marco Arkesteijn, Tracey Willis.

**Data curation:** Richa Kulshrestha, Nicholas Emery, Marco Arkesteijn, Tracey Willis.

**Formal analysis:** Fraser Philp, Richa Kulshrestha, Nicholas Emery, Marco Arkesteijn, Anand Pandyan, Tracey Willis.

**Funding acquisition:** Richa Kulshrestha, Nicholas Emery, Marco Arkesteijn, Tracey Willis.

**Investigation:** Richa Kulshrestha, Nicholas Emery, Marco Arkesteijn, Tracey Willis.

**Methodology:** Fraser Philp, Richa Kulshrestha, Nicholas Emery, Marco Arkesteijn, Tracey Willis.

**Project administration:** Fraser Philp, Richa Kulshrestha, Nicholas Emery, Marco Arkesteijn, Tracey Willis.

**Resources:** Richa Kulshrestha, Nicholas Emery, Marco Arkesteijn, Tracey Willis.

**Software:** Fraser Philp, Anand Pandyan.

**Supervision:** Richa Kulshrestha, Nicholas Emery, Marco Arkesteijn, Anand Pandyan, Tracey Willis.

**Writing – original draft:** Fraser Philp, Richa Kulshrestha.

**Writing – review & editing:** Fraser Philp, Richa Kulshrestha, Nicholas Emery, Marco Arkesteijn, Anand Pandyan, Tracey Willis.

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
