## [Decision Letter · Decision Letter 0]

14 Oct 2021

PONE-D-21-20857A pilot study of a single intermittent arm cycling exercise programme on people affected by Facioscapulohumeral dystrophy (FSHD)PLOS ONE

Dear Dr. Philp,

Thank you for submitting your manuscript to PLOS ONE. After careful consideration, we feel that it has merit but does not fully meet PLOS ONE’s publication criteria as it currently stands. Therefore, we invite you to submit a revised version of the manuscript that addresses the points raised during the review process.

The manuscript has been evaluated by two reviewers, and their comments are available below.

The reviewers feels that the study is overall well reported, however had some recommendations to further improved the manuscript.

In particular, the reviewers have requested some further details to be reported in the technical statistical section of the Methodology. And, additional clarification to the study aim is needed in light of the study variables chosen for analysis.

Could you please carefully revise the manuscript to address all comments raised?

We look forward to receiving your revised manuscript.

Kind regards,

Lucinda Shen, MSc

Staff Editor

PLOS ONE

Journal Requirements:

2. Thank you for stating the following in the Acknowledgments / Funding Section of your manuscript: 

The study was funded by The Orthopaedic Institute Limited grant reference number RPG162

The study was funded by The Orthopaedic Institute Limited grant reference number RPG162 - recipient TW 

https://www.orthopaedic-institute.org/

The funders had no role in study design, data collection and analysis, decision to publish, or preparation of the manuscript

I have read the journal's policy and the authors of this manuscript have the following competing interests: I am reporting that Professor Anand Pandyan has received unrestricted educational support from Allergan and Biometrics Ltd., and Honorarium payments from Allergan, Biometrics Ltd, Ipsen, and Merz. These companies are unlikely to be affected by the research reported in the enclosed paper. I have disclosed those interests fully and I have in place an approved plan for managing any potential conflicts arising from that involvement. 

Reviewers' comments:

Reviewer's Responses to Questions

**Comments to the Author**

1. Is the manuscript technically sound, and do the data support the conclusions?

Reviewer #1: Yes

Reviewer #2: Yes

2. Has the statistical analysis been performed appropriately and rigorously? 

Reviewer #1: Yes

Reviewer #2: Yes

3. Have the authors made all data underlying the findings in their manuscript fully available?

Reviewer #1: No

Reviewer #2: No

4. Is the manuscript presented in an intelligible fashion and written in standard English?

Reviewer #1: Yes

Reviewer #2: Yes

5. Review Comments to the Author

Reviewer #1: A pilot study was conducted to investigate the effect of an arm cycling exercise program in people affected by Facioscapulohumeral dystrophy (FSHD). Correlations were observed between strength and (a) shoulder function and (b) rate of perceived exertion of final effort against shoulder function.

Minor revisions:

1- Line 197: Identify the statistical methods used for estimating the correlations.

2- Cite the statistical software used for the analysis.

3- Lines 246-279. Provide the percentages that correspond to the frequencies.

4- Figure 5: Indicate that R-squared represents the correlation.

State the strength of the correlations according to this table:

Coefficient (absolute value) Interpretation

0.90 - 1.0 Very Strong

0.70 - 0.89 Strong

0.40 - 0.69 Moderate

0.10 - 0.39 Weak

less than 0.10 Negligible correlation

Reviewer #2: A well written exploratory, feasibility study in an area with little focused research. This is greatly needed and I hope the research team will expand this work to the trial phase.

There are some issues with this paper that need clarifying prior to being suitable for publication.

1) Please include the make of dynamometer and the type of test, e.g. make or break test

2) There needs to be more justification of the kinematic measurement. If the aim of this work was feasibility and increasing engagement in activity, what was the purpose? In the discussion you mention fatigue changing movement and compensation. If this was the purpose of the kinematic analysis, then this needs to be fully explored in the introduction (experience and potential impact of fatigue, fatigability, trick movements) and the methods (what are changes you are measuring and why).

3) What is the rationale for the intermittent programme? Dosage is not established in FSH and other muscle diseases, but I don't see why this is the exercise prescription is favoured over continuous training. Please clarify.

4) One thought I have as someone who works with this group of people is that the lower limbs are less affected in many cases so for increasing aerobic capacity, general activity and fitness, standard bicycle ergometry will allow them to reach higher work loads, more engagement and get a better training effect. I think you need to be clear about what the training goal here is. I see arm pedalling as being an opportunity to increase strength (as you state) but also muscle endurance and perhaps your prescription should reflect this. The goal of the activity needs to be clearer and linked to the underlying presentation.

5) What are the possible mechanisms of your proposed change (if effective)? Are you increasing the capacity of the compensating muscles?

6. PLOS authors have the option to publish the peer review history of their article (what does this mean?). If published, this will include your full peer review and any attached files.

Reviewer #1: No

Reviewer #2: **Yes: **Dr Gita Ramdharry

---

## [Author Response · Author response to Decision Letter 0]

18 Nov 2021

Reviewer #1: A pilot study was conducted to investigate the effect of an arm cycling exercise program in people affected by Facioscapulohumeral dystrophy (FSHD). Correlations were observed between strength and (a) shoulder function and (b) rate of perceived exertion of final effort against shoulder function.

AR - The authors would like to thank the reviewer for their time and comments regarding our manuscript. We have made the required changes or hopefully provided sufficient context regarding some of the points raised.

1- Line 197: Identify the statistical methods used for estimating the correlations.

1R - The statistical methods used for estimating correlations have been identified in the manuscript as Pearon’s correlations 

2- Cite the statistical software used for the analysis.

2R - The statistical software’s used for the analysis have been reported

3- Lines 246-279. Provide the percentages that correspond to the frequencies.

3R - The percentages that correspond to the frequencies.

The n=1 in line 279 is to represent that the references study only has a sample size of 1.

4- Figure 5: Indicate that R-squared represents the correlation.

State the strength of the correlations according to this table:

Coefficient (absolute value) Interpretation

0.90 - 1.0 Very Strong

0.70 - 0.89 Strong

0.40 - 0.69 Moderate

0.10 - 0.39 Weak

less than 0.10 Negligible correlation

4R - An annotation indicating that R2 values represent the correlation between factors has been inserted into the manuscript to support figure 5.

Thank you for your suggestion, for the coefficients we would trust the reader to take a decision on the strength of the correlations and would request not to put a value-based judgement according to the suggested coefficient value interpretations. We have therefore not stated the strength of the correlations according to the table.

Reviewer #2: A well written exploratory, feasibility study in an area with little focused research. This is greatly needed and I hope the research team will expand this work to the trial phase.

There are some issues with this paper that need clarifying prior to being suitable for publication.

AR - The authors would like to thank the reviewer for their time and comments regarding our manuscript. We were pleased to read your positive comments. We have made the required changes or hopefully provided sufficient context regarding some of the points raised

1) Please include the make of dynamometer and the type of test, e.g. make or break test

1R) - All available information regarding the make of dynamometer CITEC HHD CT3002 and the type of test (break) have been included.

2) There needs to be more justification of the kinematic measurement. If the aim of this work was feasibility and increasing engagement in activity, what was the purpose? In the discussion you mention fatigue changing movement and compensation. If this was the purpose of the kinematic analysis, then this needs to be fully explored in the introduction (experience and potential impact of fatigue, fatigability, trick movements) and the methods (what are changes you are measuring and why).

2R - Further justification regarding the use of kinematic measurement has been included in the Introduction and Materials and Methodology section.

Introduction 

“There is therefore limited evidence available to inform clinical practice and support strategies for improving physical activity and exercise for people affected by FSHD (8). This includes identification of measures e.g. body structure (e.g. movement features) or body systems responses (e.g. cardiovascular) that can be used to identify fatigue leading to adverse exercise responses or monitoring of changes in control during exercise activities. Given the limited functional capacity and secondary complications experienced in people affected by FSHD, there is a need identify alternate methods for increasing and measuring physical activity engagement that may benefit shoulder function and wider health.”

Materials and Methods

“This was done to explore if any features of fatigue or adverse responses could be identified based on participant movement profile characteristics.”

Please also see our final response to the reviewer 

3) What is the rationale for the intermittent programme? Dosage is not established in FSH and other muscle diseases, but I don't see why this is the exercise prescription is favoured over continuous training. Please clarify.

3R) The reviewer raises an important point and we have provided further clarification in the manuscript and amended as below

“This exercise programme type was selected as it allows for distinct periods of exercise efforts, between which, patients’ responses to exercise could be evaluated i.e. development of delayed or immediate adverse responses to exercise. As there are no guidelines specific to this exercise intervention in patients with FSHD, the intermittent programme type was a pragmatic selection allowing for consideration of patient safety and evaluation of a dose-response relationship to exercise.”

4) One thought I have as someone who works with this group of people is that the lower limbs are less affected in many cases so for increasing aerobic capacity, general activity and fitness, standard bicycle ergometry will allow them to reach higher work loads, more engagement and get a better training effect. I think you need to be clear about what the training goal here is. I see arm pedalling as being an opportunity to increase strength (as you state) but also muscle endurance and perhaps your prescription should reflect this. The goal of the activity needs to be clearer and linked to the underlying presentation.

4R) Again, the reviewer raises an interesting and important point. We would agree with the reviewer here and have included further elaboration in the introduction 

This is important as a majority of the impairments in patients with FSHD occur in the upper-limb, limiting activities dependant on upper-limb function and participation. 

And discussion

“Alternate exercise programme variables (e.g. intensity, time, body structure (lower versus upper limb)) may be used to achieve specific or concurrent responses to exercise (e.g. endurance, strength, general physical activity) on the basis of first principle arguments or clinical reasoning, which maybe reflective of current clinical practice. However, these need to be considered against the previously discussed points.”

We acknowledge the wider points made by the reviewer re consideration of alternative exercise methods / outcome aims but feel this discussion is beyond the scope of our study. Whilst agreeably correct and interesting discussion topic there is limited available evidence and the aim of our study was to provide some pilot data which may contribute to this boy of work.

5) What are the possible mechanisms of your proposed change (if effective)? Are you increasing the capacity of the compensating muscles?

5R) We acknowledge the point made by the reviewer. At this stage we feel it is too early to say and we are currently undertaking/ planning future studies to investigate this further.

We hypothesise the primary aim would be to prevent deterioration given the natural history of the condition.

Engagement with the exercise intervention may increase the capacity of the existing muscles / synergies and potentially increase the functional capacity through conditioning of the redundant/ compensatory muscles which are used throughout the disease progression. 

Currently however we don’t have the evidence for this and would prefer not to commit to a position as of yet.

---

## [Decision Letter · Decision Letter 1]

13 May 2022

A pilot study of a single intermittent arm cycling exercise programme on people affected by Facioscapulohumeral dystrophy (FSHD)

PONE-D-21-20857R1

Dear Dr. Philp,

We’re pleased to inform you that your manuscript has been judged scientifically suitable for publication and will be formally accepted for publication once it meets all outstanding technical requirements.

Please see my comments below for a minor change that should be made prior to the manuscript's publication. Please accept our apologies for any inconvenience caused by this.

Kind regards,

George Vousden

Deputy Editor in Chief

PLOS ONE

Additional Editor Comments (optional):

In the previous round of review, Reviewer 1 requested that the caption of Figure 5 should indicate that the R-squared value represents the correlation. However, as the review notes below, R-squared is actually the coefficient of determination. Before your manuscript is sent your production please either A) remove the text "R2 values represent the
correlation between factors" from the figure legend or B) indicate that R2 values represent the coefficient of determination. 

Reviewers' comments:

Reviewer's Responses to Questions

**Comments to the Author**

1. If the authors have adequately addressed your comments raised in a previous round of review and you feel that this manuscript is now acceptable for publication, you may indicate that here to bypass the “Comments to the Author” section, enter your conflict of interest statement in the “Confidential to Editor” section, and submit your "Accept" recommendation.

Reviewer #1: (No Response)

2. Is the manuscript technically sound, and do the data support the conclusions?

Reviewer #1: Yes

3. Has the statistical analysis been performed appropriately and rigorously? 

Reviewer #1: Yes

4. Have the authors made all data underlying the findings in their manuscript fully available?

Reviewer #1: Yes

5. Is the manuscript presented in an intelligible fashion and written in standard English?

Reviewer #1: Yes

6. Review Comments to the Author

Reviewer #1: Pertaining to the graphs, R-squared is actually the coefficient of determination. In fact, it's the square of Pearson's correlation coefficient. Sorry for the confusion.

7. PLOS authors have the option to publish the peer review history of their article (what does this mean?). If published, this will include your full peer review and any attached files.

Reviewer #1: No

---

## [Editor Report · Acceptance letter]

14 Jun 2022

PONE-D-21-20857R1 

A pilot study of a single intermittent arm cycling exercise programme on people affected by Facioscapulohumeral dystrophy (FSHD). 

Dear Dr. Philp:

I'm pleased to inform you that your manuscript has been deemed suitable for publication in PLOS ONE. Congratulations! Your manuscript is now with our production department. 

Kind regards, 

on behalf of

Dr. George Vousden 

Staff Editor

PLOS ONE